# A lytic transglycosylase connects bacterial focal adhesion complexes to the peptidoglycan cell wall

Carlos A Ramirez Carbo[1,2†], Olalekan G Faromiki[1†], Beiyan Nan[1*]

[1]Department of Biology, Texas A&M University, College Station, United States; [2]The Genetics and Genomics Interdisciplinary Program, Texas A&M University, College Station, United States

*For correspondence:
bnan@bio.tamu.edu

†These authors contributed equally to this work

Competing interest: The authors declare that no competing interests exist.

**Abstract** The Gram-negative bacterium *Myxococcus xanthus* glides on solid surfaces. Dynamic bacterial focal adhesion complexes (bFACs) convert proton motive force from the inner membrane into mechanical propulsion on the cell surface. It is unclear how the mechanical force transmits across the rigid peptidoglycan (PG) cell wall. Here, we show that AgmT, a highly abundant lytic PG transglycosylase homologous to *Escherichia coli* MltG, couples bFACs to PG. Coprecipitation assay and single-particle microscopy reveal that the gliding motors fail to connect to PG and thus are unable to assemble into bFACs in the absence of an active AgmT. Heterologous expression of *E. coli* MltG restores the connection between PG and bFACs and thus rescues gliding motility in the *M. xanthus* cells that lack AgmT. Our results indicate that bFACs anchor to AgmT-modified PG to transmit mechanical force across the PG cell wall.

## eLife assessment

The manuscript by Ramirez Carbo et al. reports a novel role for the MltG homolog AgmT in gliding motility in *M. xanthus*. The authors provide **convincing** data to demonstrate that AgmT is a cell wall lytic enzyme (likely a lytic transglycosylase), its lytic activity is required for gliding motility, and that its activity is required for proper binding of a component of the motility apparatus to the cell wall. The findings are **valuable** as they contribute to our understanding of the molecular mechanisms underlying the interaction between gliding motility and the bacterial cell wall.

## Introduction

In natural ecosystems, the majority of bacteria attach to surfaces (*Laventie and Jenal, 2020*). Surface-associated motility is critical for many bacteria to navigate and populate their environments (*Nan and Zusman, 2016*). The Gram-negative bacterium *Myxococcus xanthus* moves on solid surfaces using two independent mechanisms: social (S-) motility and adventurous (A-) motility. S-motility, analogous to the twitching motility in *Pseudomonas* and *Neisseria*, is powered by the extension and retraction of type IV pili (*Chang et al., 2016*; *Wu and Kaiser, 1995*). A-motility is a form of gliding motility that does not depend on conventional motility-related cell surface appendages, such as flagella or pili (*Nan and Zusman, 2016*; *Nan and Zusman, 2011*). AglR, AglQ, and AglS form a membrane channel that functions as the gliding motor by harvesting proton motive force (*Nan et al., 2011*; *Sun et al., 2011*). Motor units associate with at least 14 gliding-related proteins that reside in the cytoplasm, inner membrane, periplasm, and outer membrane (*Jakobczak et al., 2015*; *Nan et al., 2010*).

In each gliding cell, two distinct populations of gliding complexes coexist: dynamic complexes move along helical paths and static complexes remain fixed relative to the substrate (*Faure et al.,*

**Figure 1.** Stationary bacterial focal adhesion complexes (bFACs) drive *M. xanthus* gliding. Motors carrying incomplete gliding complexes either diffuse or move rapidly along helical paths but do not generate propulsion. Motors stall and become nearly static relative to the substrate when they assemble into complete bFACs with other motor-associated proteins at the ventral side of the cell. Stalled motors push MreB and bFACs in opposite directions and thus exert force against outer membrane adhesins. Overall, as motors transport bFACs toward lagging cell poles, cells move forward but bFACs remain static relative to the substrate. IM, inner membrane; OM, outer membrane.

*2016*; *Nan et al., 2013*; *Nan et al., 2014*; *Nan, 2017*). Motors transport incomplete gliding complexes along helical tracks but form complete, force-generating complexes at the ventral side of cells that appear fixed to the substrate (*Figure 1*; *Faure et al., 2016*; *Nan, 2017*). Based on their functional analogy with eukaryotic focal adhesion sites, these complete gliding machineries are called bacterial focal adhesion complexes (bFACs). Despite their static appearance, bFACs are dynamic under single-particle microscopy. Individual motors frequently display move–stall–move patterns, in which they move rapidly along helical trajectories, join a bFAC where they remain stationary briefly, then move rapidly again toward the next bFAC or reverse their moving directions (*Nan et al., 2013*; *Nan et al., 2015*; *Pogue et al., 2018*). bFACs adhere to the gliding substrate through an outer membrane adhesin (*Islam et al., 2023*; *Nan et al., 2014*). As motors transport bFACs toward lagging cell poles, cells move forward but bFACs remain static relative to the gliding substrate.

bFAC assembly can be quantified at nanometer resolution using the single-particle dynamics of gliding motors (*Nan et al., 2013*; *Fu et al., 2018*). Single particles of a fully functional, photoactivatable mCherry (PAmCherry)-labeled AglR display three dynamic patterns, stationary, directed motion, and diffusion (*Nan et al., 2013*; *Fu et al., 2018*). The stationary population consists of the motors in fully assembled bFACs, which do not move before photobleach. In contrast, the motors moving in a directed manner carry incomplete gliding complexes along helical tracks, whereas the diffusing motors assemble to even less completeness (*Faure et al., 2016*; *Nan et al., 2013*; *Nan, 2017*; *Fu et al., 2018*). As motors only generate force in static bFACs (*Faure et al., 2016*), the population of nonmotile motors indicates the overall status of fully assembled bFACs.

However, how bFACs transmit force across the peptidoglycan (PG) cell wall is unclear. PG is a mesh-like single molecule of crosslinked glycan strands that surrounds the entire cytoplasmic membrane. The rigidity of PG defines cell shape and protects cells from osmotic lysis (*Garner, 2021*; *Rohs and Bernhardt, 2021*). If bFACs physically penetrate PG, their transportation toward lagging cell poles would tear PG and trigger cell lysis. To avoid breaching PG, an updated model proposes that rather than forming stable and rigid complexes, gliding-related proteins only assemble into force-generating machineries in bFACs. Outside of bFACs, these proteins could localize diffusively or move along with unengaged motor units (*Faure et al., 2016*; *Chen and Nan, 2022*). To simplify this model, we can artificially divide each gliding machinery into two subcomplexes, one on each side of the PG layer. The inner subcomplexes move freely and only assemble with the outer subcomplexes in bFACs, which transform proton motive force into mechanical propulsion on cell surfaces (*Figure 1*).

The inner subcomplexes, containing the motors, are the force-generating units in bFACs. As motors reside in the fluid inner membrane, to transmit force to the cell surface, the inner subcomplexes must push against two relatively rigid structures, one on each side of the membrane, in opposite directions (*Chen and Nan, 2022*; *Figure 1*). MreB is a bacterial actin homolog that supports rod shape in many bacteria (*Fu et al., 2018*; *Garner, 2021*; *Zhang et al., 2021*). In *M. xanthus*, MreB also connects to bFACs on the cytoplasmic side and thus plays essential roles in gliding (*Nan and Zusman, 2016*; *Nan and Zusman, 2011*; *Nan et al., 2011*; *Nan et al., 2014*; *Fu et al., 2018*; *Mauriello et al., 2010*; *Treuner-Lange et al., 2015*). The inner subcomplexes could push against MreB filaments and PG in the cytoplasm and periplasm, respectively (*Faure et al., 2016*; *Fu et al., 2018*; *Chen and Nan, 2022*; *Zhang et al., 2020*). The interaction between the inner subcomplexes and PG not only satisfies the physical requirement for force generation but also supports bFAC stability. Without this interaction, the inner and outer subcomplexes can only form transient, 'slippery' association, which is predicted to produce short and aberrant cell movements (*Chen and Nan, 2022*). How the inner complex interacts with PG remains unknown. It was speculated that gliding motors in the inner complex could bind PG directly (*Faure et al., 2016*). However, such binding has not been confirmed by experiments.

In this study, we found that AgmT, a lytic transglycosylase (LTG) for PG, is required for *M. xanthus* gliding motility. Whereas AgmT only regulates cell morphology moderately during vegetative growth, it is essential for maintaining PG integrity under the stress from the antibiotic mecillinam. Using single-particle tracking microscopy and coprecipitation assays, we found that AgmT is essential for the inner subcomplexes to connect to PG and stall in bFACs. Importantly, expressing *Escherichia coli* MltG heterologously rescues the connection between PG and bFACs and thus restores gliding motility. Hence, the LTG activity of AgmT anchors bFACs to PG, potentially by modifying PG structure. Our findings reveal the long-sought connection between PG and bFACs that allows mechanical force to transmit across the PG cell wall.

## Results
### AgmT, a putative LTG, is required for gliding motility

To elucidate how bFACs interact with PG, we searched for potential PG-binding domains among the proteins that are required for gliding motility. A previous report identified 35 gliding-related genes in *M. xanthus* through transposon-mediated random mutagenesis (*Youderian et al., 2003*). Among these genes, *agmT* (ORF K1515_04910, MXAN_6607 *Aramayo and Nan, 2022*) was predicted to encode an inner membrane protein with a single transmembrane helix (residues 4–25) followed by a large 'periplasmic solute-binding' domain (*Youderian et al., 2003*). No other motility-related genes are found in the vicinity of *agmT*. After careful analysis, we found that AgmT showed significant similarity to the widely conserved YceG/MltG family LTGs. The putative active site, Glu223 (corresponding to E218 in *E. coli* MltG) (*Yunck et al., 2016*), is conserved in AgmT (*Figure 2A*).

To confirm the function of AgmT in gliding, we constructed an *agmT* in-frame deletion mutant. We further knocked out the *pilA* gene that encodes pilin for type IV pilus to eliminate S-motility. On a 1.5% agar surface, the *pilA⁻* cells moved away from colony edges both as individuals and in 'flare-like' cell groups, indicating that they were still motile with gliding motility. In contrast, the *ΔaglR pilA⁻* cells that lack an essential component in the gliding motor were unable to move outward and thus formed sharp colony edges. Similarly, the *ΔagmT pilA⁻* cells also formed sharp colony edges, indicating that they could not move efficiently with gliding (*Figure 2B*).

We then imaged individual *ΔagmT pilA⁻* cells on a 1.5% agar surface at 10-s intervals using bright-field microscopy. To our surprise, instead of being static, individual *ΔagmT pilA⁻* cells displayed slow movements, with frequent pauses and reversals (*Video 1*). To quantify the effects of AgmT, we measured the velocity and gliding persistency (the distances cells traveled before pauses and reversals) of individual cells. Compared to the *pilA⁻* cells that moved at 2.30 ± 1.33 µm/min (*n* = 46) and high persistency (*Video 2* and *Figure 2C*, *Figure 2—source data 1*, *Figure 2D*, *Figure 2—source data 2*), *ΔagmT pilA⁻* cells moved significantly slower (0.88 ± 0.62 µm/min, *n* = 59) and less persistent (*Video 1* and *Figure 2C*, *Figure 2—source data 1*, *Figure 2D*, *Figure 2—source data 2*). Such aberrant gliding motility is distinct from the 'hyper reversal' phenotype. Although the hyper reversing cells constitutively switch their moving directions, they usually maintain gliding velocity at the wild-type level (*Leonardy et al., 2010*). Instead, the slow and 'slippery' gliding of the *ΔagmT pilA⁻* cells matches

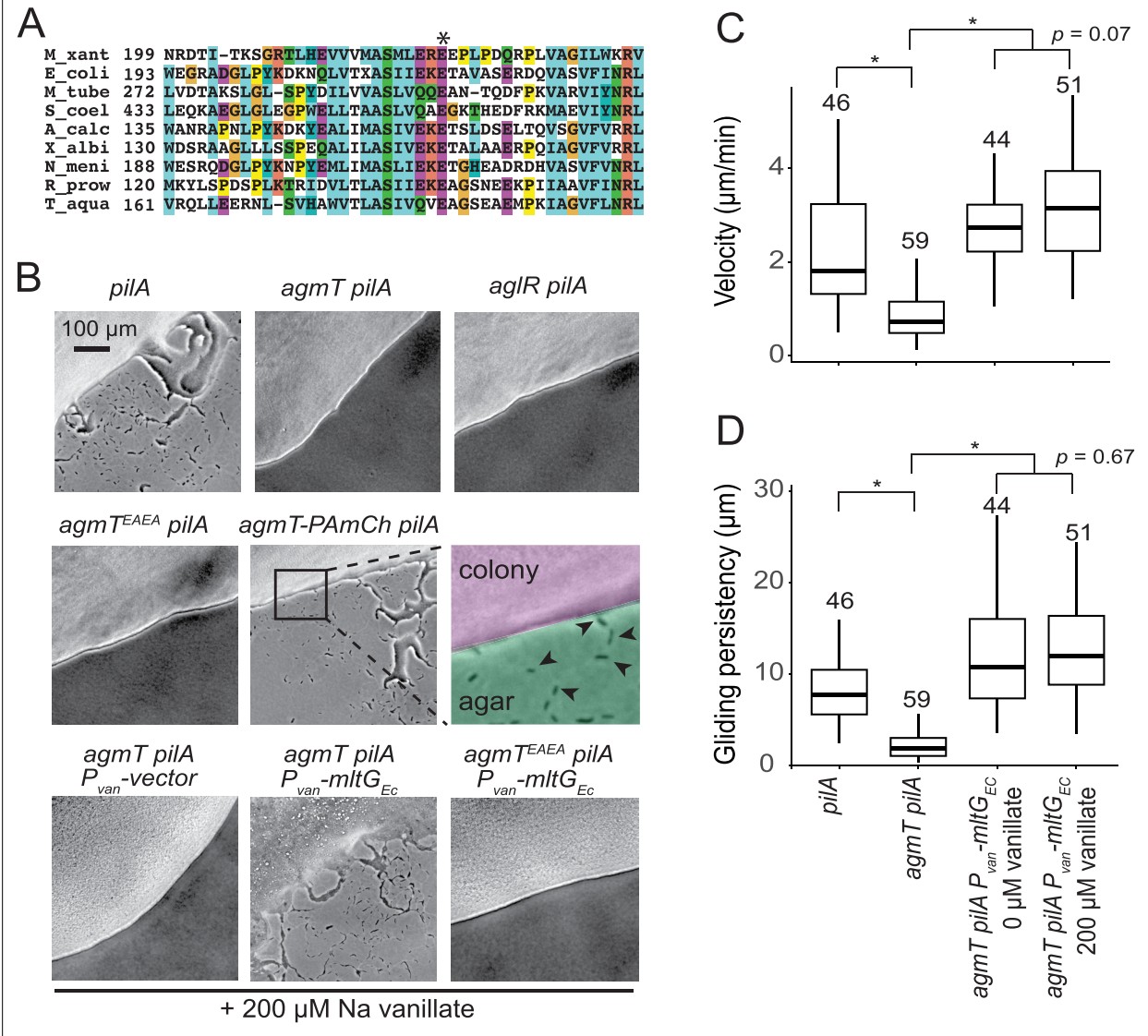

**Figure 2.** AgmT, a putative lytic transglycosylase, is required *for M. xanthus* gliding. (**A**) AgmT shows significant similarity to a widely conserved peptidoglycan (PG) transglycosylase in the YceG/MltG family. The conserved glutamine residue is marked by an asterisk. M_xant, *M. xanthus*; E_coli, *E. coli*; M_tube, *Mycobacterium tuberculosis*; S_coel, *Streptomyces coelicolor*; A_calc, *Acinetobacter calcoaceticus*; X_albi, *Xanthomonas albilineans*; N_meni, *Neisseria menningitidis*; R_prow, *Rickettsia prowazekii*; T_aqua, *Thermus aquaticus*. (**B**) AgmT is required for *M. xanthus* gliding. Colony edges were imaged after incubating cells on 1.5% agar surfaces for 24 hr. To eliminate S-motility, we further knocked out the *pilA* gene that encodes pilin for type IV pilus. Cells that move by gliding are able to move away from colony edges (as pointed by the arrows in the inset). Deleting *agmT* or disabling the active site of AgmT (AgmT$^{EAEA}$) abolish gliding but fusing an PAmCherry (PAmCh) to its C-terminus does not. Heterologous Expression of *E. coli* MltG (MltG$_{Ec}$) restores gliding of *agmT* cells but not the cells that express AgmT$^{EAEA}$. While cells lacking AgmT moved slower (**C**) and less persistently (**D**, measured by the distances cells traveled before pauses and reversals), the expression of MltG$_{Ec}$ restores both the velocity and persistency of gliding in the *agmT* cells. Data were pooled from three biological replicates and p values were calculated using a one-way analysis of variance (ANOVA) test between two unweighted, independent samples. Boxes indicate the 25th to 75th percentiles and bars the median. The total number of cells analyzed is shown on top of each plot. *p < 0.001. Growth curves of *agmT*-related mutants can be found in *Figure 2—figure supplement 1*. Other putative lytic transglycosylases (LTGs) do not affect gliding motility significantly, which is shown in *Figure 2—figure supplement 2*.

The online version of this article includes the following source data and figure supplement(s) for figure 2:

**Source data 1.** Gliding velocity data for *Figure 2C*.

**Source data 2.** Gliding persistency data for *Figure 2D*.

**Figure supplement 1.** AgmT does not regulate growth.

**Figure supplement 2.** Other putative lytic transglycosylases (LTGs) are not required for *M. xanthus* gliding motility.

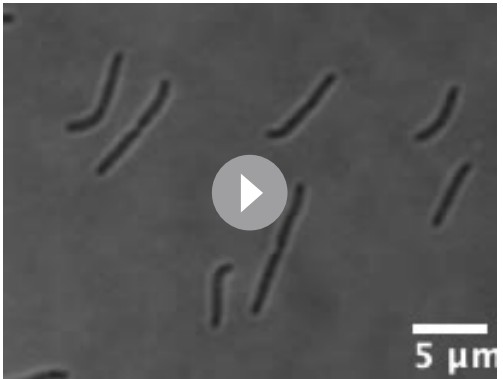

**Video 1.** Gliding motility of the Δ*agmT pilA⁻* cells. Timelapse was captured at 10-s intervals and the video plays at 10 frames/s (100× speedup).

https://elifesciences.org/articles/99273/figures#video1

the prediction that when the inner complexes of bFACs lose connection with PG, bFACs can only generate short, and inefficient movements (*Chen and Nan, 2022*). Our data indicate that AgmT is not an essential component in the bFACs. Thus, AgmT is likely to regulate the assembly and stability of bFACs, especially their connection with PG.

## AgmT is an LTG

As AgmT is a putative LTG for PG, we then tested if its predicted active site, Glu223, is required for gliding. Because the amino acid following Glu223 is also a glutamate (Glu224), we replaced both glutamate residues with alanine (AgmT$^{EAEA}$) using site-directed mutagenesis to alter their codons on the chromosome. *agmT$^{EAEA}$ pilA⁻* cells formed sharp colony edges on agar surface (*Figure 2B*). Thus, the putative LTG activity of AgmT is required

for *M. xanthus* gliding motility.

To determine if AgmT is an LTG, we expressed the periplasmic domains (amino acids 25–339) of wild-type AgmT and AgmT$^{EAEA}$ in *E. coli*. We purified PG from wild-type *M. xanthus* cells, labeled it with Remazol brilliant blue (RBB), and tested if the purified AgmT variants hydrolyze labeled PG in vitro and release the dye (*Uehara et al., 2010*; *Jorgenson et al., 2014*). Similar to lysozyme that specifically cleaves the β-1,4-glycosidic bonds in PG, wild-type AgmT solubilized dye-labeled *M. xanthus* PG that absorbed light at 595 nm. In contrast, the AgmT$^{EAEA}$ variant failed to release the dye (*Figure 3A*, *Figure 3—source data 1*). Hence, AgmT displays LTG activity in vitro.

Whereas AgmT does not affect growth rate (*Figure 2—figure supplement 1*), cells that lacked AgmT or expressed AgmT$^{EAEA}$ maintained rod shape but were slightly shorter and wider than the wild-type ones (*Figure 3B*, *Figure 3—source data 2*). Nevertheless, altered morphology alone does not likely account for abolished gliding. In fact, a previously reported mutant that lacks all three class A penicillin-binding proteins (*Δ3*) displays similarly shortened and widened morphology (*Zhang et al., 2023*) but is still motile by gliding (*Figure 3—figure supplement 1*, *Figure 3—source data 2*).

A recent report revealed that *Vibrio cholerae* MltG degrades un-crosslinked PG turnover products and prevents their detrimental accumulation in the periplasm (*Weaver et al., 2022*). To test if AgmT plays a similar role in *M. xanthus*, we used mecillinam to induce cell envelope stress. Mecillinam is a β-lactam that induces the production of toxic, un-crosslinked PG strands (*Cho et al., 2014*). Rather than collapsing the rod shape, mecillinam only causes bulging near the centers of wild-type *M. xanthus* cells, whereas large-scale cell lysis does not occur, and cell poles still maintain rod shape even after prolonged (20 hr) treatment (*Figure 3C*; *Zhang et al., 2023*). Thus, wild-type *M. xanthus* can largely mitigate mecillinam stress. In contrast, mecillinam-treated *agmT* cells increased their width drastically, displayed significant cell surface irregularity along their entire cell bodies, and lysed frequently (*Figure 3C*). Cells expressing AgmT$^{EAEA}$ as the sole source of AgmT displayed similar, but even stronger phenotypes under mecillinam stress, growing into elongated, twisted filaments that formed multiple cell poles and lysed frequently (*Figure 3C*). These results confirm that similar to *V. cholerae* MltG, *M. xanthus* AgmT is important for maintaining cell integrity under mecillinam stress. The different

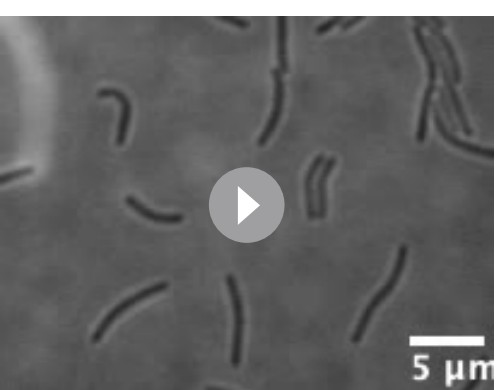

**Video 2.** Gliding motility of the *pilA⁻* cells. Timelapse was captured at 10-s intervals and the video plays at 10 frames/s (100× speedup).

https://elifesciences.org/articles/99273/figures#video2

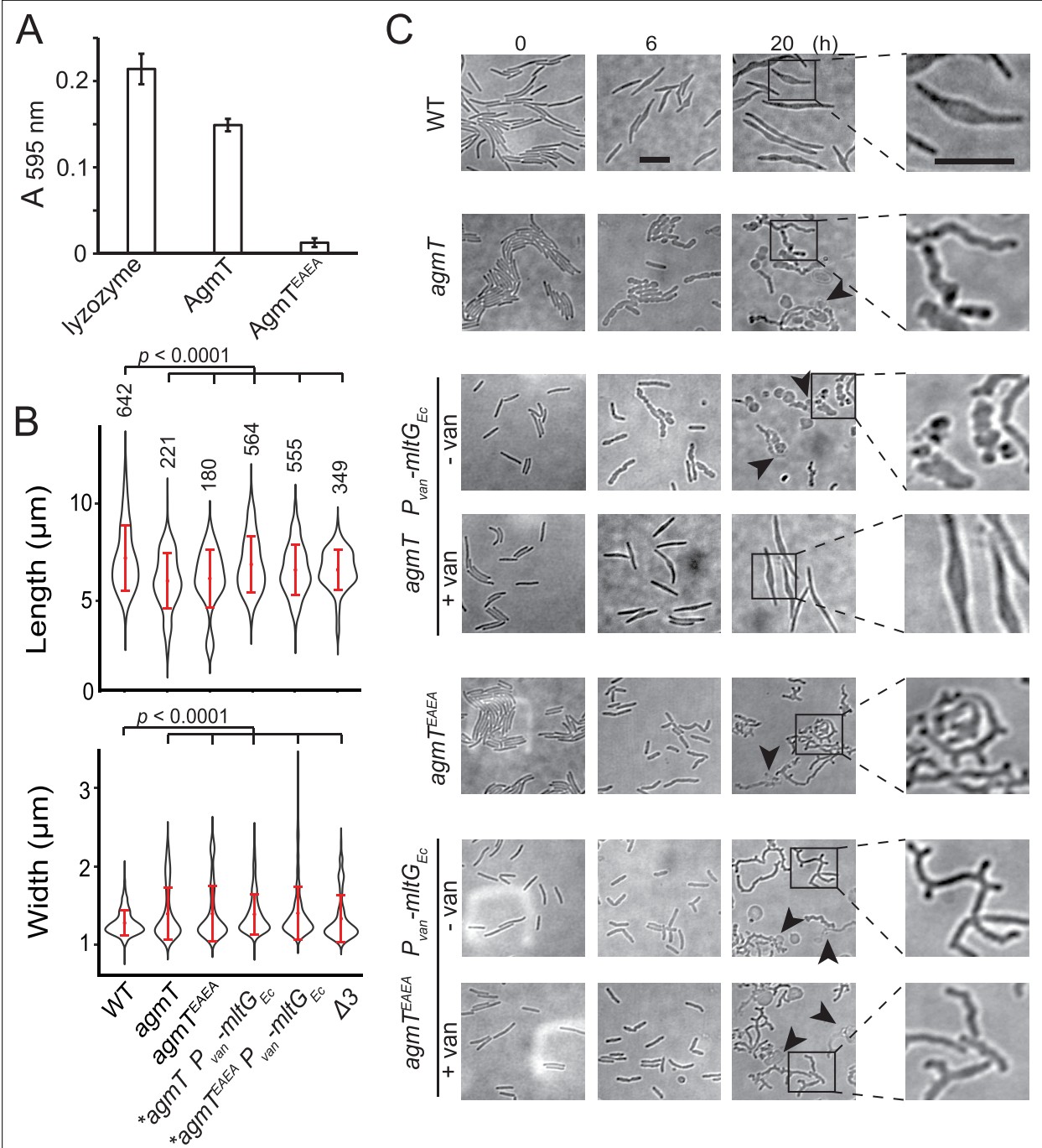

**Figure 3.** AgmT regulates cell morphology and integrity under antibiotic stress. (**A**) Purified AgmT solubilizes dye-labeled peptidoglycan (PG) sacculi, but AgmT$^{EAEA}$ does not. Lysozyme serves as a positive control. Absorption at 595 nm was measured after 18 hr incubation at 25°C. Data are presented as mean values ± standard deviation (SD) from three technical replicates. (**B**) AgmT regulates cell morphology. Compared to wild-type cells, cells that lack AgmT and express AgmT$^{EAEA}$ are significantly shorter and wider. A previously reported mutant that lacks all three class A penicillin-binding proteins (*Δ3*) displays similarly shortened and widened morphology but is still motile by gliding (*Figure 3—figure supplement 1*). Heterologous expression of *E. coli* MltG partially restores cell length but not cell width in *agmT* and *agmT$^{EAEA}$* backgrounds. Asterisks, 200 μM sodium vanillate added. Data were pooled from two biological replicates and p values were calculated using a one-way ANOVA test between two unweighted, independent samples. Whiskers indicate the 25th to 75th percentiles and red dots the median. The total number of cells analyzed is shown on top of each plot. (**C**) AgmT regulates cell morphology and integrity under mecillinam stress (100 μg/ml). Expressing *E. coli* MltG by a vanillate-inducible promoter ($P_{van}$) restores resistance against mecillinam in *agmT* cells but not in the cells that express AgmT$^{EAEA}$. van, 200 μM sodium vanillate. Arrows point to newly lysed cells. Scale bars, 5 μm.

The online version of this article includes the following source data and figure supplement(s) for figure 3:

*Figure 3 continued on next page*

*Figure 3 continued*

**Source data 1.** Absorption at 595 nm in the Remazol brilliant blue (RBB) assay for *Figure 3C*.

**Source data 2.** Cell morphology (cell length and width) data for *Figure 3D*.

**Figure supplement 1.** Moderate changes in cell dimensions do not affect gliding motility significantly.

responses from *agmT* and *agmT^EAEA* cells suggest that while other LTGs could partially substitute AgmT, AgmT^EAEA blocks these enzymes from accessing un-crosslinked PG strands.

AgmT is the only LTG *in M. xanthus* that belongs to the YceG/MltG family. Besides *agmT*, the genome of *M. xanthus* contains 13 genes that encode putative LTGs (Key resources table). To test if these proteins also contribute to gliding, we knocked out each of these 13 genes in the *pilA^−* background. The resulting mutants all retained gliding motility, indicating that AgmT is the only LTG that is required for *M. xanthus* gliding (*Figure 2—figure supplement 2*).

## AgmT is essential for proper bFACs assembly

How does AgmT support gliding? We first tested if AgmT regulates the function of the gliding motor. For this purpose, we quantified the motor dynamics by tracking single particles of AglR, an essential component of the motor. We spotted the cells that express a fully functional, photo-activatable mCherry (PAmCherry)-labeled AglR (*Nan et al., 2013*) on 1.5% agar surfaces. We used a 405-nm

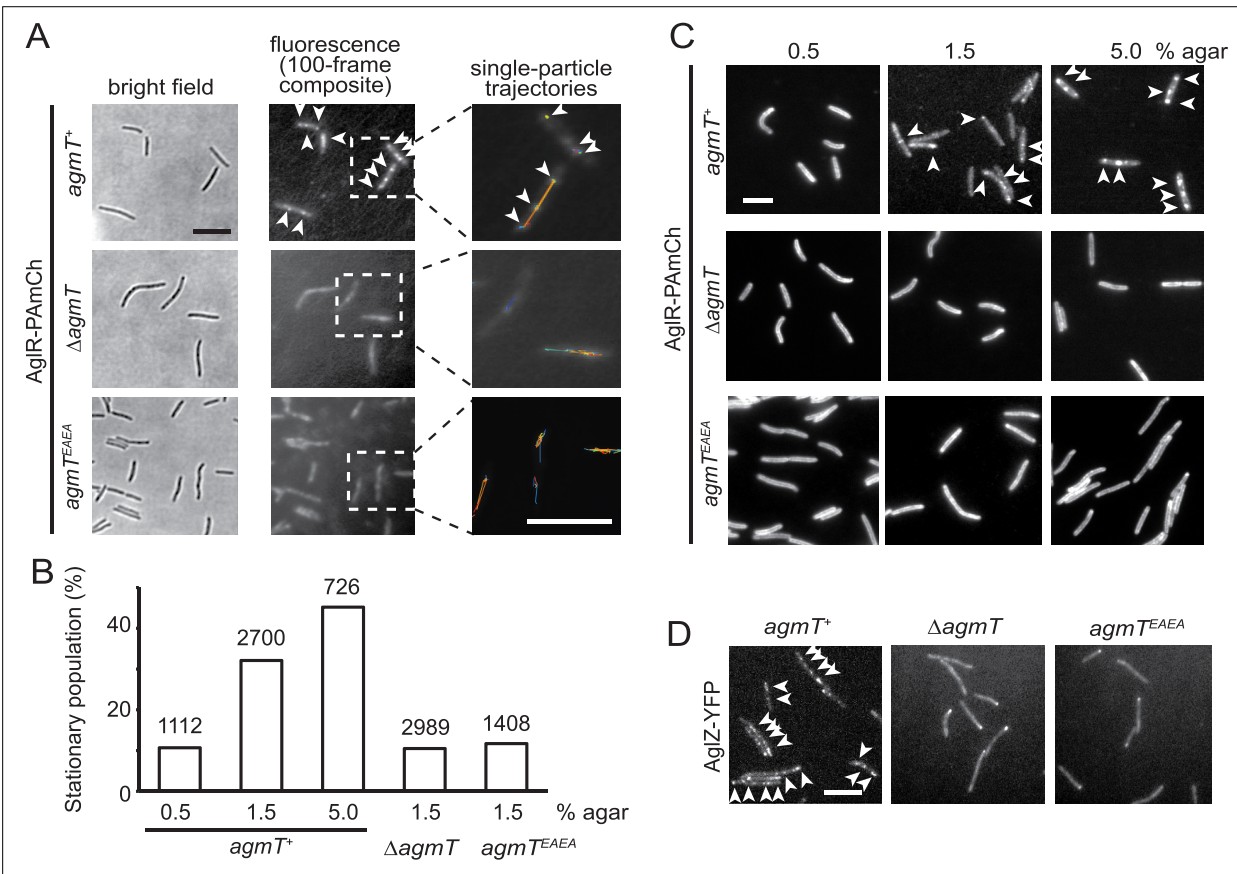

**Figure 4.** AgmT and its lytic transglycosylase (LTG) activity are essential for proper bacterial focal adhesion complex (bFAC) assembly. (**A**) Overall distribution of AglR-PAmCherry particles on 1.5% agar surface is displayed using the composite of 100 consecutive frames taken at 100-ms intervals. Single-particle trajectories of AglR-PAmCherry (AglR-PAmCh) were generated from the same frames. Individual trajectories are distinguished by colors. Deleting *agmT* or disabling the transglycosylase activity of AgmT (AgmT^EAEA) decrease the stationary population of AglR particles. (**B**) The stationary population of AglR-PAmCherry particles, which reflects the AglR molecules in bFACs, changes in response to substrate hardness (controlled by agar concentration) and the presence and function of AgmT. The total number of particles analyzed is shown on top of each plot. (**C**) AglR fails to assemble into bFACs in the absence of active AgmT, even on 5.0% agar surfaces. (**D**) AgmT and its LTG activity also support the assembly of AglZ into bFACs. White arrows point to bFACs. Scale bars, 5 μm.

excitation laser to activate the fluorescence of a few labeled AglR particles randomly in each cell and quantified their localization using a 561-nm laser at 10 Hz using single-particle tracking photo-activated localization microscopy (sptPALM) under highly inclined and laminated optical sheet (HILO) illumination (*Nan et al., 2013*; *Fu et al., 2018*; *Tokunaga et al., 2008*). Using this setting, only a thin section of each cell surface that was close to the coverslip was illuminated. To analyze the data, we only chose the fluorescent particles that remained in focus for 4–12 frames (0.4–1.2 s). As free PAmCherry particles diffuse extremely fast in the cytoplasm, entering and exiting the focal plane frequently, they usually appear as blurry objects that cannot be followed at 10 Hz close to the membrane (*Fu et al., 2018*). For this reason, the noise from any potential degradation of AglR-PAmCherry was negligible. Consistent with our previous results (*Nan et al., 2013*; *Nan et al., 2015*), 32.1% (*n* = 2700) of AglR-PAmCherry particles remained within one pixel (160 nm × 160 nm) before photobleach, indicating that they were immotile. The remaining 67.9% AglR-PAmCherry particles were motile, leaving trajectories of various lengths (*Figure 4A, B*).

bFAC assembly is sensitive to mechanical cues. As the agar concentration increases in gliding substrate, more motor molecules engage in bFACs, where they appear immotile (*Nan et al., 2010*; *Nan et al., 2013*). We exposed *aglR-PAmCherry* cells to 405 nm excitation (0.2 kW/cm$^2$) for 2 s, where most PAmCherry molecules were photoactivated and used epifluorescence to display the overall localization of AglR (*Nan et al., 2013*; *Fu et al., 2018*; *Zhang et al., 2023*). As the agar concentration increased, bFACs increased significantly in size (*Figure 4C*). Accordingly, the immotile AglR particles detected by sptPALM increased from 10.7% (*n* = 1112) on 0.5% agar to 45.2% (*n* = 726) on 5.0% agar surface (*Figure 4B*). Thus, the stationary population of AglR-PAmCherry particles reflects the AglR molecules in fully assembled bFACs.

In contrast to the cells that expressed wild-type AgmT (*agmT*$^+$), AglR particles were hyper motile in the cells that lacked AgmT or expressed AgmT$^{EAEA}$. On 1.5% agar surfaces, the immotile populations of AglR particles decreased to 10.5% (*n* = 2989) and 11.7% (*n* = 1408), respectively. Consistently, AglR-containing bFACs were rarely detectable in these cells, even on 5% agar surfaces (*Figure 4A–C*).

To test if other components in the gliding machinery also depend on AgmT to assemble into bFACs, we tested the localization of AglZ, a cytoplasmic protein that is commonly used for assessing bFAC assembly (*Faure et al., 2016*; *Mignot et al., 2007*). In the cells that expressed wild-type AgmT, yellow fluorescent protein (YFP)-labeled AglZ formed a bright cluster at the leading cell pole and multiple static, near-evenly spaced clusters along the cell body, indicating the assembly of bFACs (*Figure 4D*). In contrast, AglZ-YFP only formed single clusters at cell poles in the cells that lacked AgmT or expressed AgmT$^{EAEA}$ (*Figure 4D*). Taken together, the LTG activity of AgmT is essential for proper bFACs assembly.

## AgmT does not assemble into bFACs

We first hypothesized that AgmT could assemble into bFACs, where it could generate pores in PG specifically at bFAC loci that allow the inner and outer gliding complexes to interact directly. Alternatively, it could bind to PG and recruit other components to bFACs through protein–protein interactions. Regardless, if AgmT assembles into bFACs, it should localize in bFACs. To test this possibility, we labeled AgmT with PAmCherry on its C-terminus and expressed the fusion protein as the sole source of AgmT using the native promoter and locus of *agmT*. *agmT-PAmCherry pilA*$^-$ cells displayed gliding motility that was indistinguishable from *pilA*$^-$ cells (*Figure 2B*), indicating that the fusion protein is functional. Similar to many membrane proteins that are resistant to dissociation by sodium dodecyl sulfate (SDS) (*Rath et al., 2009*), immunoblot using an anti-mCherry antibody showed that AgmT-PAmCherry accumulated in two bands in SDS–polyacrylamide gel electrophoresis (PAGE) that corresponded to monomers and dimers of the full-length fusion protein, respectively (*Figure 5A*, *Figure 5—source data 1 and 2*). This result is consistent with the structure of *E. coli* MltG that functions as homodimers (PDB: 2r1f). Importantly, AgmT-PAmCherry was about 50 times more abundant than AglR-PAmCherry (*Figure 5A*, *Figure 5—source data 1 and 2*). To test if AgmT assembles into bFACs, we first expressed AgmT-PAmCherry and AglZ-YFP together using their respective loci and promoter. We exposed *agmT-PAmCherry* cells to 405 nm excitation (0.2 kW/cm$^2$) for 2 s to visualize the overall localization of AgmT. We found that on a 1.5% agar surface that favors gliding motility, AglZ formed bright clusters at cell poles and aggregated in near-evenly spaced bFACs along the cell body. In contrast, AgmT localized near evenly along cell bodies without forming protein clusters (*Figure 5B*).

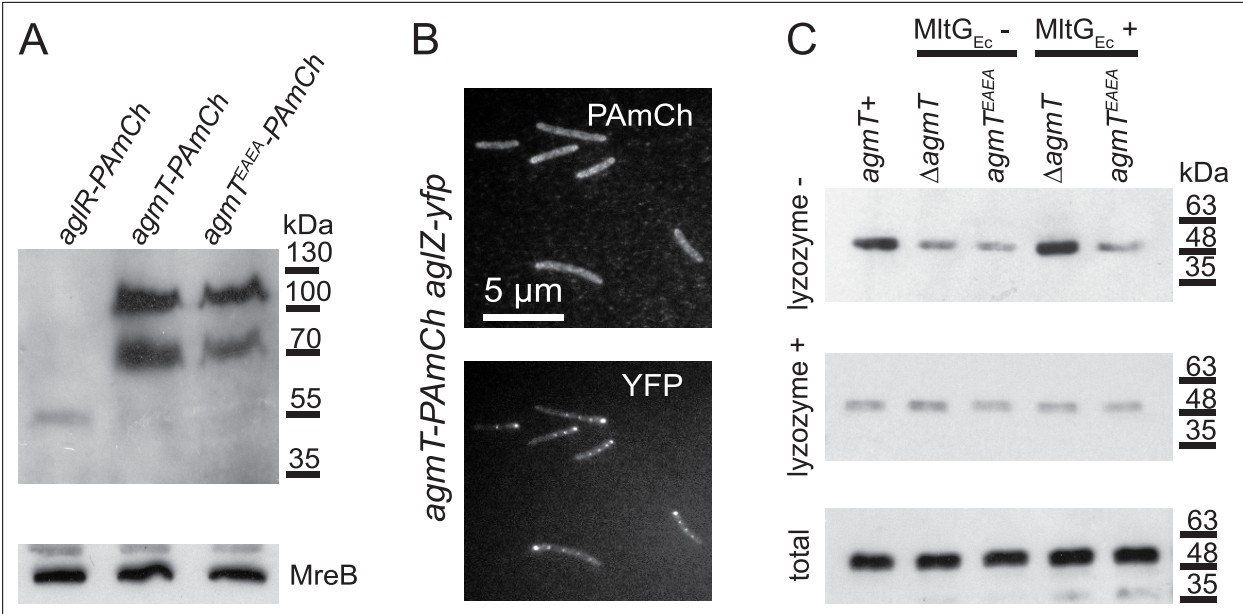

**Figure 5.** AgmT does not assemble into bacterial focal adhesion complexes (bFACs) but connects bFACs to peptidoglycan (PG). (**A**) Immunoblotting using *M. xanthus* cell lysates and an anti-mCherry antibody shows that PAmCherry (PAmCh)-labeled AgmT and AgmT^EAEA (592 amino acids) accumulate as full-length proteins. AgmT is significantly more abundant than the PAmCherry-labeled motor protein AglR (498 amino acids). The bacterial actin homolog MreB visualized using an MreB antibody is shown as a loading control. (**B**) AgmT does not aggregate into bFACs. (**C**) The lytic transglycosylase (LTG) activity of AgmT is required for connecting bFACs to PG and expressing the *E. coli* LTG MltG (MltG_Ec) restores PG binding by bFACs in cells that lack AgmT but not in the ones that express an inactive AgmT variant (AgmT^EAEA). AglR-PAmCherry was detected using an anti-mCherry antibody to mark the presence of bFACs that co-precipitate with PG-containing (lysozyme−) pellets. Lysates from the cells that express AglR-PAmCherry in different genetic backgrounds were pelleted by centrifugation in the presence and absence of lysozyme. The loading control of AglR-PAmCherry in the whole cell is shown as 'total'.

The online version of this article includes the following source data for figure 5:

**Source data 1.** The original file of the full, raw, and unedited blot for the upper panel of *Figure 5A*.

**Source data 2.** The original file of the full, raw, and unedited blot for the lower panel of *Figure 5A*.

**Source data 3.** The original file of the full, raw, and unedited blot for the upper panel of *Figure 5C*.

**Source data 4.** The original file of the full, raw, and unedited blot for the middle panel of *Figure 5C*.

**Source data 5.** The original file of the full, raw, and unedited blot for the lower panel of *Figure 5C*.

Thus, AgmT does not localize into bFACs. These results echo the fact that despite its abundance, AgmT has not been identified as a component in bFACs despite extensive pull-down experiments using various bFAC components as baits (*Nan et al., 2011*; *Nan et al., 2010*; *Jolivet et al., 2023*).

As an additional test, we used sptPALM to track the movements of AgmT-PAmCherry single particles on 1.5% agar surfaces at 10 Hz. We reasoned that if AgmT assembles into bFACs, AgmT and gliding motors should display similar dynamic patterns. Distinct from the AglR particles of which 32.1% remained stationary, AgmT moved in a diffusive manner, showing no significant immotile population. Importantly, compared to the diffusion coefficients ($D$) of motile AglR particles ($1.8 \times 10^{-2} \pm 3.6 \times 10^{-3}$ µm²/s ($n = 1833$)), AgmT particles diffused much faster ($D = 2.9 \times 10^{-2} \pm 5.3 \times 10^{-3}$ µm²/s ($n = 8548$)). Taken together, AgmT and the gliding motor did not display significant correlation in either their localization or dynamics. We thus conclude that AgmT does not assemble into bFACs.

## AgmT connects bFACs to PG through its LTG activity

We then explored the possibility that AgmT could modify PG through its LTG activity and thus generate the anchor sites for certain components in bFACs. If this hypothesis is true, heterologous expression of a non-native LTG could rescue gliding motility in the *ΔagmT* strain. To test this hypothesis, we fused the *E. coli mltG* gene (*mltG_Ec*) to a vanillate-inducible promoter and inserted the resulting construct to the *cuoA* locus on *M. xanthus* chromosome that does not interfere gliding (*Iniesta et al., 2012*). We subjected the *ΔagmT* and *agmT^EAEA* cells that expressed MltG_Ec to mecillinam (100 µg/ml) stress.

Induced by 200 μM sodium vanillate, MltG$_{Ec}$ restored cell morphology and integrity of the $\Delta agmT$ strain to the wild-type level (*Figure 3C*). This result confirmed that MltG$_{Ec}$ was expressed and enzymatically active in *M. xanthus*. In contrast, MltG$_{Ec}$ failed to confer mecillinam resistance to the *agmT*$^{EAEA}$ cells (*Figure 3C*). A potential explanation is that AgmT$^{EAEA}$ could still bind to PG and thus block MltG$_{Ec}$ from accessing *M. xanthus* PG.

Consistent with its LTG activity, the expression of MltG$_{Ec}$ restored gliding motility of the $\Delta agmT$ *pilA*$^-$ cells on both the colony (*Figure 2B*) and single-cell (*Figure 2C*, *Figure 2—source data 1*, *Figure 2D*, *Figure 2—source data 2*) levels. Interestingly, in the absence of sodium vanillate, the leakage expression of MltG$_{Ec}$ using the vanillate-inducible promoter was sufficient to compensate the loss of AgmT. A plausible explanation of this observation is that as *E. coli* grows much faster (generation time 20–30 min) than *M. xanthus* (generation time ~4 hr), MltG$_{Ec}$ could possess significantly higher LTG activity than AgmT. Induced by 200 μM sodium vanillate, the expression of MltG$_{Ec}$ further but non-significantly increased the velocity and gliding persistency (*Figure 2B, C*, *Figure 2—source data 1*, *Figure 2D*, *Figure 2—source data 2*). Importantly, the expression of MltG$_{Ec}$ failed to restore gliding motility in the *agmT*$^{EAEA}$ *pilA* cells, even in the presence of 200 μM sodium vanillate (*Figure 2B*). Consistent with the mecillinam resistance assay (*Figure 3C*), this result suggests that AgmT$^{EAEA}$ still binds to PG and that in the absence of its LTG activity, AgmT does not anchor bFACs to PG. As MltG$_{Ec}$ substitutes AgmT in gliding motility, it is the LTG activity, rather than specific interactions between AgmT and bFACs, that is required for gliding.

It is challenging to visualize how AgmT facilitates bFAC assembly through its enzymatic activity in vitro. First, the inner complex in a bFAC spans the cytoplasm, inner membrane, and periplasm and contains multiple proteins whose activities are interdependent (*Jakobczak et al., 2015*; *Nan et al., 2010*; *Faure et al., 2016*). It is hence difficult to purify the entire inner complex in its functional state. Second, *Faure et al., 2016* hypothesized that AglQ and AglS in the gliding motor could bind PG directly. However, such binding has not been proved experimentally due to the difficulty in purifying the membrane integral AglRQS complex. To overcome these difficulties, we used AglR-PAmCherry to represent the inner complex of bFAC and investigated how bFACs bind PG in their native environment. To do so, we expressed AglR-PAmCherry in different genetic backgrounds as the sole source of AglR using the native *aglR* locus and promoter. We lysed these cells using sonication, subjected their lysates to centrifugation, isolated the pellets that contained PG, and detected the presence of AglR-PAmCherry in these pellets by immunoblots using an mCherry antibody. Eliminating AgmT and disabling its active site significantly reduced the amounts of AglR in the pellets (*Figure 5C*, *Figure 5—source data 3–5*). Because such pellets contain both the PG and membrane fractions, we further eliminated PG in cell lysates using lysozyme before centrifugation and determined the amount of AglR-PAmCherry in the pellets that only contained membrane fractions. Regardless of the presence and activity of AgmT, comparable amounts of AglR precipitated in centrifugation pellets after lysozyme treatment, indicating that AgmT does not affect the expression or stability of AglR (*Figure 5C*, *Figure 5—source data 3–5*). Thus, active AgmT facilitates the association between bFACs and PG. Strikingly, the heterologous expression of MltG$_{Ec}$ enriched AglR-PAmCherry in the PG-containing pellets from the $\Delta agmT$ cells but not the *agmT*$^{EAEA}$ ones (*Figure 5C*, *Figure 5—source data 3–5*). These results indicate that AgmT connects bFACs to PG through its LTG activity.

The assembly of bFACs produces wave-like deformation on cell surface (*Nan et al., 2011*; *Tchoufag et al., 2019*,) suggesting that their assembly may require a flexible PG layer (*Nan and Zusman, 2016*; *Nan et al., 2011*; *Nan et al., 2013*; *Nan et al., 2014*). As a major contributor to cell stiffness, PG flexibility affects the overall stiffness of cells (*Auer et al., 2016*). To test the possibility that AgmT and MltG$_{Ec}$ facilitate bFAC assembly by reducing PG stiffness, we adopted the general regulators affecting bacterial stiffness (GRABS) assay (*Auer et al., 2016*) to quantify if the lack of AgmT and the expression of MltG$_{Ec}$ affects cell stiffness. To quantify changes in cell stiffness, we simultaneously measured the growth of the *pilA*$^-$, $\Delta agmT$ *pilA*$^-$, and $\Delta agmT$ $P_{van}$-MltG$_{Ec}$ *pilA*$^-$ (with 200 μM sodium vanillate) cells in a 1% agarose gel infused with CYE and liquid CYE and calculated the GRABS scores of the $\Delta agmT$ *pilA*$^-$, and $\Delta agmT$ $P_{van}$-MltG$_{Ec}$ *pilA*$^-$ cells using the *pilA*$^-$ cells as the reference, where positive and negative GRABS scores indicate increased and decreased stiffness, respectively (see Materials and methods; *Auer et al., 2016*). The GRABS scores of the $\Delta agmT$ *pilA*$^-$, and $\Delta agmT$ $P_{van}$-MltG$_{Ec}$ *pilA*$^-$ (with 200 μM sodium vanillate) cells were $-0.06 \pm 0.04$ and $-0.10 \pm 0.07$ ($n = 4$), respectively, indicating that neither AgmT nor MltG$_{Ec}$ affects cell stiffness significantly. Whereas PG flexibility could still be essential for

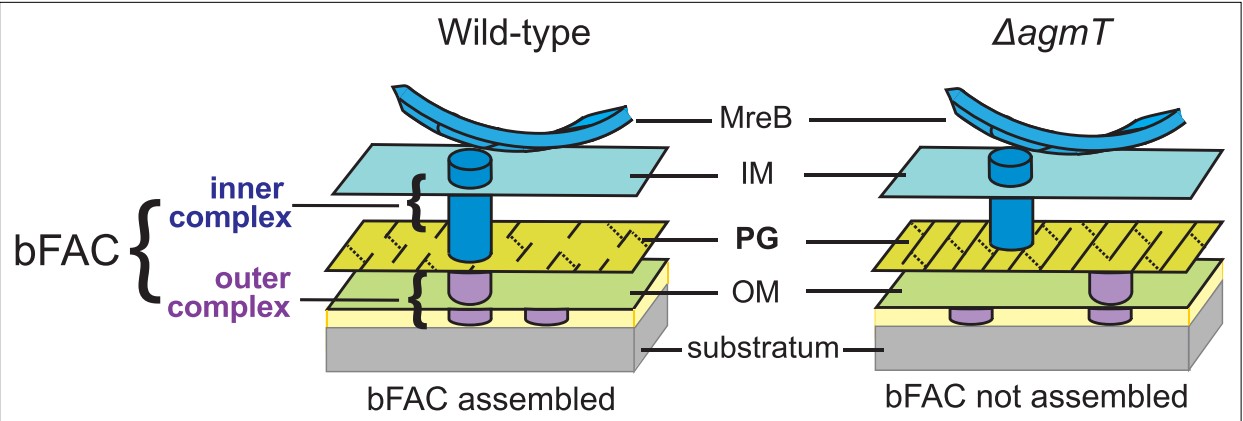

**Figure 6.** A possible mechanism by which AgmT connect bFACs to PG. AgmT could generate short glycan strands through its lytic transglycosylase (LTG) activity and thus uniquely modify the overall structure of *M. xanthus* PG, such as producing small pores that retard and retain the inner subcomplexes of bFACs. Likewise, the *M. xanthus* mutants that lack active AgmT could produce PG with increased strand length, which precludes bFACs from binding to the cell wall.

gliding, AgmT and MltG$_{Ec}$ do not regulate bFAC assembly by modulating PG stiffness. Instead, these LTGs could connect bFACs to PG by generating structural features that are irrelevant to PG stiffness.

## Discussion

Gliding bacteria adopt a broad spectrum of nanomachineries. Compared to the trans-envelope secretion system that drives gliding in *Flavobacterium johnsoniae* and *Capnocytophaga gingivalis*, the gliding machinery of *M. xanthus* appears to lack a stable structure that transverses the cell envelope, especially across the PG layer (*Nan and Zusman, 2016*; *Hennell James et al., 2021*; *Shrivastava et al., 2018*). In order to generate mechanical force from the fluid *M. xanthus* gliding machinery, the inner complex must establish solid contact with PG and push the outer complex to slide (*Faure et al., 2016*; *Chen and Nan, 2022*). In this work, we discovered AgmT as the long-sought factor that facilitates persistent gliding by connecting bFACs to PG.

It is surprising that AgmT itself does not assemble into bFACs and that MltG$_{Ec}$ substitutes AgmT in gliding. Thus, rather than interacting with bFAC components directly and specifically, AgmT facilitates proper bFAC assembly indirectly through its LTG activity. LTGs usually break glycan strands and produce unique anhydro caps on their ends (*Williams et al., 2018*; *Dik et al., 2017*; *Höltje et al., 1975*; *van Heijenoort, 2011*; *Weaver et al., 2023*). However, because AgmT is the only LTGs that is required for gliding, it is not likely to facilitate bFAC assembly by generating such modification on glycan strands. *E. coli* MltG is a glycan terminase that controls the length of newly synthesized PG glycans (*Yunck et al., 2016*). Likewise, AgmT could generate short glycan strands and thus uniquely modify the overall structure of *M. xanthus* PG, such as producing small pores that retard and retain the inner subcomplexes of bFACs (*Figure 6*). On the contrary, the *M. xanthus* mutants that lack active AgmT could produce PG with increased strand length, which blocks bFACs from binding to the cell wall and precludes stable bFAC assembly. However, it would be very difficult to demonstrate how glycan length affects the connection between bFACs and PG.

Then how does AgmT, a protein localized diffusively, facilitate bFAC assembly into near-evenly spaced foci? The actin homolog MreB is the only protein in bFACs that displays quasiperiodic localization independent of other components (*Fu et al., 2018*; *Mauriello et al., 2010*). *M. xanthus* MreB assembles into bFACs and positions the latter near evenly along the cell body (*Nan et al., 2011*; *Nan et al., 2013*; *Fu et al., 2018*; *Mauriello et al., 2010*; *Treuner-Lange et al., 2015*; *Zhang et al., 2020*). MreB filaments change their orientation in accordance with local membrane curvatures and could hence respond to mechanical cues (*Hussain et al., 2018*; *Wong et al., 2019*; *Ursell et al., 2014*). Strikingly, *M. xanthus* does assemble bFACs in response to substrate hardness (*Figure 4C*) and the assembled bFACs could distort the cell envelope, generating undulations on the cell surface (*Nan et al., 2011*; *Nan et al., 2010*; *Nan et al., 2013*; *Tchoufag et al., 2019*; *Chen et al., 2023*). Taken

together, whereas AgmT potentially modifies the entire PG layer, it is still the quasiperiodic MreB filaments that determine the loci for bFAC assembly in response to the mechanical cues from the gliding substrate (*Figure 6*).

Most bacteria encode multiple LTGs that function in PG growth, remodeling, and recycling (*Dik et al., 2017*; *Weaver et al., 2023*). This work provides an example that macro bacterial machineries domesticate non-specialized LTGs for specialized functions. Other examples include MltD in the *Helicobacter pylori* flagellum and MltE in the *E. coli* type VI secretion system (*Roure et al., 2012*; *Santin and Cascales, 2017*). Based on their catalytic folds and domain arrangements, LTGs can be categorized into six distinct families (*Dik et al., 2017*). Both *M. xanthus* AgmT and *E. coli* MltG belong to the YceG/MltG family, which is the first identified LTG family that is conserved in both Gram-negative and -positive bacteria (*Yunck et al., 2016*; *Dik et al., 2017*). About 70% of bacterial genomes, including firmicutes, proteobacteria, and actinobacteria, encode YceG/MltG domains (*Yunck et al., 2016*). The unique inner membrane localization of this family and the fact that AgmT is the only *M. xanthus* LTG that belongs to this family (Key resources table) could partially explain why it is the only LTG that contributes to gliding motility. It will be interesting to investigate if other LTGs, once anchored to the inner membrane, could also facilitate force generation by bFACs.

## Methods

### Key resources table

| Reagent type (species) or resource | Designation | Source or reference | Identifiers | Additional information |
|---|---|---|---|---|
| Gene (*M. xanthus*) | *agmT* (K1515_04910) | *Aramayo and Nan, 2022* | MXAN_RS31980 MXAN_6607 | MltG/YceG LTG family 5 |
| Gene (*M. xanthus*) | K1515_37860 | *Aramayo and Nan, 2022* | MXAN_RS00570 MXAN_0114 | Slt/MltE LTG family 1 |
| Gene (*M. xanthus*) | K1515_37385 | *Aramayo and Nan, 2022* | MXAN_RS01035 MXAN_0210 | Slt/MltE LTG family 1 |
| Gene (*M. xanthus*) | K1515_34725 | *Aramayo and Nan, 2022* | MXAN_RS03640 MXAN_0754 | LTG family unknown |
| Gene (*M. xanthus*) | K1515_24775 | *Aramayo and Nan, 2022* | MXAN_RS12365 MXAN_2558 | RlpA LTG family 6 |
| Gene (*M. xanthus*) | K1515_22185 | *Aramayo and Nan, 2022* | MXAN_RS14935 MXAN_3081 | Slt/MltE LTG family 1 |
| Gene (*M. xanthus*) | K1515_20905 | *Aramayo and Nan, 2022* | MXAN_RS16205 MXAN_3344 | Slt/MltE LTG family 1 |
| Gene (*M. xanthus*) | K1515_20820 | *Aramayo and Nan, 2022* | MXAN_RS16290 MXAN_3363 | Slt/MltE LTG family 1 |
| Gene (*M. xanthus*) | K1515_17460 | *Aramayo and Nan, 2022* | MXAN_RS19615 MXAN_4034 | Slt/MltE LTG family 1 |
| Gene (*M. xanthus*) | K1515_14545 | *Aramayo and Nan, 2022* | MXAN_RS22465 MXAN_4628 | Slt/MltE LTG family 1 |
| Gene (*M. xanthus*) | K1515_09355 | *Aramayo and Nan, 2022* | MXAN_RS27580 MXAN_5690 | Slt/MltD/MltE, LTG family 1 |
| Gene (*M. xanthus*) | K1515_06035 | *Aramayo and Nan, 2022* | MXAN_RS30855 MXAN_6370 | Slt/MltE LTG family 1 |
| Gene (*M. xanthus*) | K1515_01490 | *Aramayo and Nan, 2022* | MXAN_RS35365 MXAN_7308 | MltA LTG family 2 |
| Gene (*M. xanthus*) | K1515_01440 | *Aramayo and Nan, 2022* | MXAN_RS35415 MXAN_7318 | Slt/MltD/MltE, LTG family 1 |
| Gene (*E. coli*) | *mltG* | UniProtKB | ec:4.2.2.29 | |

*Continued on next page*

*Continued*

| Reagent type (species) or resource | Designation | Source or reference | Identifiers | Additional information |
|---|---|---|---|---|
| Strain, strain background (*M. xanthus*) | DZ2 | *Campos and Zusman, 1975* | DZ2 | Wild-type strain |
| Strain, strain background (*M. xanthus*) | aglR-PAmCherry | *Nan et al., 2013* | N/A | |
| Strain, strain background (*M. xanthus*) | Δ3 (Δpbp1a1 Δpbp1a2 pbp1c::kan) | *Zhang et al., 2023* | BN311 | |
| Strain, strain background (*M. xanthus*) | aglZ-yfp::kan | *Mignot et al., 2007* | TM7 | |
| Strain, strain background (*M. xanthus*) | Δ3 (Δpbp1a1 Δpbp1a2 pbp1c::kan) pilA::tet | This paper | BN328 | Request from Nan lab |
| Strain, strain background (*M. xanthus*) | ΔagmT | This paper | BN329 | Request from Nan lab |
| Strain, strain background (*M. xanthus*) | ΔagmT pilA::tet | This paper | BN330 | Request from Nan lab |
| Strain, strain background (*M. xanthus*) | agmT$^{EAEA}$ | This paper | BN331 | Request from Nan lab |
| Strain, strain background (*M. xanthus*) | agmT$^{EAEA}$ pilA::tet | This paper | BN332 | Request from Nan lab |
| Strain, strain background (*M. xanthus*) | ΔaglR pilA::tet | This paper | BN333 | Request from Nan lab |
| Strain, strain background (*M. xanthus*) | agmT-PAmCherry | This paper | BN334 | Request from Nan lab |
| Strain, strain background (*M. xanthus*) | agmT$^{EAEA}$-PAmCherry | This paper | BN335 | Request from Nan lab |
| Strain, strain background (*M. xanthus*) | agmT-PAmCherry pilA::tet | This paper | BN336 | Request from Nan lab |
| Strain, strain background (*M. xanthus*) | ΔagmT pilA::tet pMR3679 | This paper | BN337 | Request from Nan lab |
| Strain, strain background (*M. xanthus*) | ΔagmT pMR3679-mltG$_{EC}$ | This paper | BN338 | Request from Nan lab |
| Strain, strain background (*M. xanthus*) | ΔagmT pMR3679-mltG$_{EC}$ pilA::tet | This paper | BN339 | Request from Nan lab |
| Strain, strain background (*M. xanthus*) | agmT$^{EAEA}$ pMR3679-mltG$_{EC}$ | This paper | BN340 | Request from Nan lab |
| Strain, strain background (*M. xanthus*) | agmT$^{EAEA}$ pMR3679-mltG$_{EC}$ pilA::tet | This paper | BN341 | Request from Nan lab |
| Strain, strain background (*M. xanthus*) | aglR-PAmCherry ΔagmT | This paper | BN342 | Request from Nan lab |
| Strain, strain background (*M. xanthus*) | aglR-PAmCherry agmT$^{EAEA}$ | This paper | BN343 | Request from Nan lab |
| Strain, strain background (*M. xanthus*) | aglZ-yfp::kan ΔagmT | This paper | BN344 | Request from Nan lab |
| Strain, strain background (*M. xanthus*) | aglZ-yfp::kan agmT$^{EAEA}$ | This paper | BN345 | Request from Nan lab |
| Strain, strain background (*M. xanthus*) | aglR-PAmCherry aglZ-yfp::kan | This paper | BN346 | Request from Nan lab |
| Strain, strain background (*M. xanthus*) | K1515_37860::kan pilA::tet | This paper | BN347 | Request from Nan lab |

*Continued on next page*

*Continued*

| Reagent type (species) or resource | Designation | Source or reference | Identifiers | Additional information |
|---|---|---|---|---|
| Strain, strain background (*M. xanthus*) | *K1515_37385::kan pilA::tet* | This paper | BN348 | Request from Nan lab |
| Strain, strain background (*M. xanthus*) | *K1515_34725::kan pilA::tet* | This paper | BN349 | Request from Nan lab |
| Strain, strain background (*M. xanthus*) | *ΔK1515_24775 pilA::tet* | This paper | BN350 | Request from Nan lab |
| Strain, strain background (*M. xanthus*) | *K1515_22185::kan pilA::tet* | This paper | BN351 | Request from Nan lab |
| Strain, strain background (*M. xanthus*) | *K1515_20905::kan pilA::tet* | This paper | BN352 | Request from Nan lab |
| Strain, strain background (*M. xanthus*) | *K1515_20820::kan pilA::tet* | This paper | BN353 | Request from Nan lab |
| Strain, strain background (*M. xanthus*) | *K1515_17460::kan pilA::tet* | This paper | BN354 | Request from Nan lab |
| Strain, strain background (*M. xanthus*) | *K1515_14545::kan pilA::tet* | This paper | BN355 | Request from Nan lab |
| Strain, strain background (*M. xanthus*) | *K1515_09355::kan pilA::tet* | This paper | BN356 | Request from Nan lab |
| Strain, strain background (*M. xanthus*) | *K1515_06035::kan pilA::tet* | This paper | BN357 | Request from Nan lab |
| Strain, strain background (*M. xanthus*) | *K1515_01490::kan pilA::tet* | This paper | BN358 | Request from Nan lab |
| Strain, strain background (*M. xanthus*) | *K1515_01440::kan pilA::tet* | This paper | BN359 | Request from Nan lab |
| Antibody (Anti-mCherry) | Rabbit polyclonal | Rockland Immunochemicals | 600-401-P16 | (1:2000) |
| Antibody (Anti-MreB) | Rabbit polyclonal | *Mauriello et al., 2010* | | (1:20,000) |
| Antibody (anti-rabbit IgG H+L (HRP)) | Goat polyclonal | Fisher Scientific | 31460 | (1:20,000) |
| Software, algorithm | MATLAB | MathWorks | | |
| Software, algorithm | Cell morphology analysis algorithm | *Zhang et al., 2020*; *Zhang et al., 2023* | | DOI: 10.5281/zenodo.8234126 |
| Software, algorithm | Single-particle analysis algorithm | *Fu et al., 2018*; *Zhang et al., 2023* | | DOI: 10.5281/zenodo.8234126 |
| Software, algorithm | TrackMate 7 | *Ershov et al., 2022* | | |
| Software, algorithm | ImageJ | https://imagej.net | | |

## Strains and growth conditions

*M. xanthus* strains used in this study are listed in Key resources table. Newly created strains are available upon request to the corresponding author. Vegetative *M. xanthus* cells were grown in liquid CYE medium (10 mM 3-(N-morpholino)propanesulfonic acid (MOPS) pH 7.6, 1% (wt/vol) Bacto casitone (BD Biosciences), 0.5% yeast extract and 8 mM MgSO$_4$) at 32°C, in 125 ml flasks with vigorous shaking, or on CYE plates that contains 1.5% agar. All genetic modifications on *M. xanthus* were made on the chromosome. Deletion and insertion mutants were constructed by electroporating *M. xanthus* cells with 4 µg of plasmid DNA. Transformed cells were plated on CYE plates supplemented with 100 µg/ml sodium kanamycin sulfate and 10 µg/ml tetracycline hydrochloride when needed. AgmT and AgmT$^{EAEA}$ were labeled with PAmCherry at their C-termini by fusing their gene to a DNA sequence that encodes PAmCherry through a KESGSVSSEQLAQFRSLD (AAGGAGTCCGGCTCCGTGTCCTCC GAGCAGCTGGCCCAGTTCCGCTCCCTGGAC) linker. All constructs were confirmed by polymerase chain reaction (PCR) and DNA sequencing.

## Immunoblotting

The expression and stability of PAmCherry-labeled proteins were determined by immunoblotting following SDS–PAGE using an anti-mCherry antibody (Rockland Immunochemicals, Inc, Lot 46705) and a goat anti-Rabbit IgG (H+L) secondary antibody, horseradish peroxidase (HRP) (Thermo Fisher Scientific, catalog # 31460). MreB was detected as the loading control using an anti-MreB serum (*Mauriello et al., 2010*) and the same secondary antibody. The blots were developed with Pierce ECL Western Blotting Substrate (Thermo Fisher Scientific REF 32109) and a MINI-MED 90 processor (AFP Manufacturing).

## Gliding assay

Five microliters of cells from overnight culture were spotted on CYE plates containing 1.5% agar at $4 \times 10^9$ colony formation units (cfu)/ml and incubated at 32°C for 48 hr. Colony edges were photographed using a Nikon Eclipse e600 phase-contrast microscope with a 10× 0.30 NA objective and an OMAX A3590U camera.

## Protein expression and purification

DNA sequences encoding amino acids 25–339 of AgmT and AgmT$^{EAEA}$ were amplified by PCR and inserted into the pET28a vector (Novogen) between the restriction sites of *Eco*RI and *Hind*III and used to transform *E. coli* strain BL21(DE3). Transformed cells were cultured in 20 ml LB (Luria-Bertani) broth at 37°C overnight and used to inoculate 1 l LB medium supplemented with 1.0% glucose. Protein expression was induced by 0.1 mM isopropyl-h-D-thiogalactopyranoside when the culture reached an $OD_{600}$ of 0.8. Cultivation was continued at 16°C for 10 hr before the cells were harvested by centrifugation at $6000 \times g$ for 20 min. Proteins were purified using a NGC Chromatography System (Bio-Rad) and 5 ml HisTrap columns (Cytiva) (*Pogue et al., 2018*; *Nan et al., 2006*). Purified proteins were concentrated using Amicon Ultra centrifugal filter units (Millipore Sigma) with a 10-kDa molecular weight cutoff and stored at −80°C.

## LTG activity (RBB) assay

PG was purified following the published protocol (*Zhang et al., 2023*; *Alvarez et al., 2016*). In brief, *M. xanthus* cells were grown until mid-stationary phase and harvested by centrifugation ($6000 \times g$, 20 min, 25°C). Supernatant was discarded and the pellet was resuspended and boiled in 1× phosphate-buffered saline (PBS) with 5% SDS for 2 hr. SDS was removed by repetitive wash with water and centrifugation ($21,000 \times g$, 10 min, 25°C). Purified PG from 100 ml culture was suspended into 1 ml 1× PBS and stored at −20°C. RBB labeling of PG was performed essentially as previously described (*Uehara et al., 2010*; *Jorgenson et al., 2014*). Purified sacculi were incubated with 20 mM RBB in 0.25 M NaOH overnight at 37°C. Reactions were neutralized by adding equal volumes of 0.25 M HCl and RBB-labeled PG was collected by centrifugation at $21,000 \times g$ for 15 min. Pellets were washed repeatedly with water until the supernatants became colorless. RBB-labeled sacculi were incubated with purified AgmT and AgmT$^{EAEA}$ (1 mg/ml) at 25°C for 12 hr. Lysozyme (1 mg/ml) was used as a positive control. Dye release was quantified by the absorption at 595 nm from the supernatants after centrifugation ($21,000 \times g$, 10 min, 25°C).

## Cell stiffness (GRABS) assay

Cell stiffness was quantified using the GRABS assay (*Adan-Kubo et al., 2006*). Briefly, overnight cultures of *pilA⁻*, *ΔagmT pilA⁻*, and *ΔagmT P_{van}-MltG_{Ec} pilA⁻* (with 200 μM sodium vanillate) cells were inoculated to liquid CYE medium or embedded in solid CYE medium with 1% agarose to $OD_{600}$ 0.5 in 2 ml optical spectrometer cuvettes and incubated at 32°C. Growth data were collected over 24 hr, and the GRABS scores were calculated as $(OD_{mutant, agarose}/OD_{pilA, agarose}) - (OD_{mutant, liquid}/OD_{pilA, liquid})$.

## Co-precipitation assay

*M. xanthus* cells expressing AglR-PAmCherry were grown in liquid CYE to $OD_{600}$ ~1, harvested by centrifugation ($6000 \times g$, 20 min, 25°C), washed by 1× PBS, and resuspended into 1× PBS to $OD_{600}$ 6.0 Cells (1 ml) were lysed using a Cole-Parmer 4710 Ultrasonic Homogenizer. Unbroken cells and large debris were eliminated by centrifugation ($6000 \times g$, 10 min, 25°C). Supernatants were subjected to centrifugation at $21,000 \times g$ for 15 min. Pellets that contain both the PG and membrane fractions

were resuspended to 1 ml in 1× PBS. To collect the pellets that do not contain PG, supernatants from sonication lysates were incubated with 5 mg/ml lysozyme at 25°C for 5 hr before centrifugation. Five microliters of each resuspended pellet were mixed with 5 µl 2× loading buffer and applied to SDS–PAGE. AglR-PAmCherry was detected using immunoblotting.

## Microscopy analysis

For all imaging experiments, we spotted 5 µl of cells grown in liquid CYE medium to $OD_{600}$ ~1 on agar (1.5%) pads. For the treatments with inhibitors, inhibitors were added into both the cell suspension and agar pads. The length and width of cells were determined from differential interference contrast (DIC) images using a MATLAB (MathWorks) script (*Fu et al., 2018*; *Zhang et al., 2023*). DIC and fluorescence images of cells were captured using an Andor iXon Ultra 897 EMCCD camera (effective pixel size 160 nm) on an inverted Nikon Eclipse-Ti microscope with a 100× 1.49 NA TIRF objective. For sptPALM, *M. xanthus* cells were grown in CYE to $4 \times 10^8$ cfu/ml and PAmCherry was activated using a 405-nm laser (0.3 $kW/cm^2$), excited and imaged using a 561-nm laser (0.2 $kW/cm^2$). Images were acquired at 10 Hz. For each sptPALM experiment, single PAmCherry particles were localized in at least 100 individual cells from three biological replicates. sptPALM data were analyzed using a MATLAB (MathWorks) script (*Fu et al., 2018*; *Zhang et al., 2023*). Briefly, cells were identified using DIC images. Single PAmCherry particles inside cells were fit by a symmetric 2D Gaussian function, whose center was assumed to be the particle's position (*Fu et al., 2018*). Particles that explored areas smaller than 160 nm × 160 nm (within one pixel) in 0.4–1.2 s were considered immotile (*Fu et al., 2018*; *Zhang et al., 2023*). Sample trajectories were generated using the TrackMate (*Ershov et al., 2022*) plugin in the ImageJ suite (https://imagej.net).

## Acknowledgements

We would like to acknowledge the Department of Biology and the College of Arts and Sciences at Texas A&M University for the support on camera purchase. We thank the members of the Nan lab for helpful comments on the project.

## Additional information

### Funding

| Funder | Grant reference number | Author |
|---|---|---|
| National Institutes of Health | R01GM129000 | Carlos A Ramirez Carbo Olalekan G Faromiki Beiyan Nan |
| National Institutes of Health | T32GM135115 | Carlos A Ramirez Carbo |

The funders had no role in study design, data collection, and interpretation, or the decision to submit the work for publication.

### Author contributions

Carlos A Ramirez Carbo, Conceptualization, Data curation, Formal analysis, Investigation, Visualization, Writing – review and editing; Olalekan G Faromiki, Data curation, Formal analysis, Investigation; Beiyan Nan, Conceptualization, Resources, Data curation, Formal analysis, Supervision, Funding acquisition, Investigation, Visualization, Writing - original draft, Project administration, Writing – review and editing

### Author ORCIDs

Beiyan Nan ⓘ https://orcid.org/0000-0002-0326-9529

Reviewer #1 (Public review): https://doi.org/10.7554/eLife.99273.3.sa1
Reviewer #2 (Public review): https://doi.org/10.7554/eLife.99273.3.sa2
Author response https://doi.org/10.7554/eLife.99273.3.sa3

## Additional files

### Supplementary files
• MDAR checklist

### Data availability
All data generated or analyzed during this study are included in the manuscript and supporting files; source data files have been provided for Figures 2, 3, and 5.

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
