## [Editor Report · eLife assessment]

The manuscript by Ramirez Carbo et al. reports a novel role for the MltG homolog AgmT in gliding motility in *M. xanthus*. The authors provide **convincing** data to demonstrate that AgmT is a cell wall lytic enzyme (likely a lytic transglycosylase), its lytic activity is required for gliding motility, and that its activity is required for proper binding of a component of the motility apparatus to the cell wall. The findings are **valuable** as they contribute to our understanding of the molecular mechanisms underlying the interaction between gliding motility and the bacterial cell wall.

---

## [Referee Report · Reviewer #1 (Public review)]

Summary:

This manuscript nicely outlines a conceptual problem with the bFAC model in A-motility, namely, how the energy derived from the inner membrane AglRQS motor transduced through the cell wall into mechanical force on the cell surface to drive motility? To address this, the authors make a significant contribution by identifying and characterizing a lytic transglycosylase (LTG) called AgmT. This work thus provides clues and a future framework work to address mechanical force transmission from the cytoplasm through the cell envelope to the cell surface.

Strengths:

(i) Convincing evidence shows AgmT functions as a LTG and, surprisingly, that mltG from e shows AgmT functions as a LTG complements the swarming defect of an agmT mutant.

(ii) Show 13 other LTGs found in M. xanthus are not required for A-motility.

(iii) Authors show agmT mutants develop morphological changes in response to treatment with a beta-lactam antibiotic, mecillinam.

(iv) The use of single molecule tracking to monitor the assembly and dynamics of bFACs in WT and mutant backgrounds.

(v) The authors understand the limitations of their work and do not overinterpret their data.

Weaknesses:

The authors provided more experiments and clearly addressed my prior concerns in their revised manuscript.

---

## [Referee Report · Reviewer #2 (Public review)]

The manuscript by Carbo et al. reports a novel role for the MltG homolog AgmT in gliding motility in M. xanthus. The authors conclusively show that AgmT is a cell wall lytic enzyme (likely a lytic transglycosylase), its lytic activity is required for gliding motility, and that its activity is required for proper binding of a component of the motility apparatus to the cell wall. The data are generally well-controlled. The marked strength of the manuscript includes the detailed characterization of AgmT as a cell wall lytic enzyme, and the careful dissection of its role in motility. Using multiple lines of evidence, the authors conclusively show that AgmT does not directly associate with the motility complexes, but that instead its absence (or the overexpression of its active site mutant) results in failure of focal adhesion complexes to properly interact with the cell wall.

---

## [Author Response]

The following is the authors’ response to the original reviews.

**Public Reviews:**

**Reviewer #1 (Public Review):**
Summary:This manuscript nicely outlines a conceptual problem with the bFAC model in A-motility, namely, how is the energy produced by the inner membrane AglRQS motor transduced through the cell wall into mechanical force on the cell surface to drive motility? To address this, the authors make a significant contribution by identifying and characterizing a lytic transglycosylase (LTG) called AgmT. This work thus provides clues and a future framework work for addressing mechanical force transmission between the cytoplasm and the cell surface.Strengths:(1) Convincing evidence shows AgmT functions as an LTG and, surprisingly, that mltG from *E. coli* complements the swarming defect of an agmT mutant.(2) Authors show agmT mutants develop morphological changes in response to treatment with a b-lactam antibiotic, mecillinam.(3) The use of single-molecule tracking to monitor the assembly and dynamics of bFACs in WT and mutant backgrounds.(4) The authors understand the limitations of their work and do not overinterpret their data.Weaknesses:(1) A clear model of AgmT's role in gliding motility or interactions with other A-motility proteins is not provided. Instead, speculative roles for how AgmT enzymatic activity could facilitate bFAC function in A-motility are discussed.

We appreciate the reviewer for this comment. We have added a new figure, Fig. 6, and updated the Discussion to propose a mechanism, “rather than interacting with bFAC components directly and specifically, AgmT facilitates proper bFAC assembly indirectly through its LTG activity. LTGs usually break glycan strands and produce unique anhydro caps on their ends40-44. However, because AgmT is the only LTGs that is required for gliding, it is not likely to facilitate bFAC assembly by generating such modification on glycan strands. *E. coli* MltG is a glycan terminase that controls the length of newly synthesized PG glycans25. Likewise, AgmT could generate short glycan strands and thus uniquely modify the overall structure of *M. xanthus* PG, such as producing small pores that retard and retain the inner subcomplexes of bFACs (Fig. 6). On the contrary, the *M. xanthus* mutants that lack active AgmT could produce PG with increased strain length, which blocks bFACs from binding to the cell wall and precludes stable bFAC assembly. However, it would be very difficult to demonstrate how glycan length affects the connection between bFACs and PG”.

(2) Although agmT mutants do not swarm, in-depth phenotypic analysis is lacking. In particular, do individual agmT mutant cells move, as found with other swarming defective mutants, or are agmT mutants completely nonmotile, as are motor mutants?

We appreciate the reviewer for bringing up an important question. Prompted by this question, we analyzed the gliding phenotype of the *ΔagmT pilA* mutant on the single cell level. We found that the *ΔagmT pilA* cells are not completely static. Instead, they move for less than half cell length before pauses and reversal. We moved on to quantify the velocity and gliding persistency and found that the gliding phenotype of the *ΔagmT pilA* cells matches the prediction on the bFACs that loses the connection between the inner subcomplexes and PG.

We then imaged individual *∆agmT pilA-* cells on 1.5% agar surface at 10-s intervals using bright-field microscopy. To our surprise, instead of being static, individual *∆agmT pilA-* cells displayed slow movements, with frequent pauses and reversals (Video 1). To quantify the effects of AgmT, we measured the velocity and gliding persistency (the distances cells traveled before pauses and reversals) of individual cells. Compared to the *pilA-* cells that moved at 2.30 ± 1.33 μm/min (n = 46) and high persistency (Video 2 and Fig. 2C, D), *∆agmT pilA-* cells moved significantly slower (0.88 ± 0.62 μm/min, n = 59) and less persistent (Video 1 and Figure. 2C, D). Such aberrant gliding motility is distinct from the “hyper reversal” phenotype. Although the hyper reversing cells constitutively switching their moving directions, they usually maintain gliding velocity at the wild-type level27. due to the polarity regulators Instead, the slow and “slippery” gliding of the *∆agmT pilA-* cells matches the prediction that when the inner complexes of bFACs lose connection with PG, bFACs can only generate short, and inefficient movements19. Our data indicate that AgmT is not essential component in the bFACs. Thus, AgmT is likely to regulate the assembly and stability of bFACs, especially their connection with PG.

(3) The bioinformatic and comparative genomics analysis of agmT is incomplete. For example, the sequence relationships between AgmT, MltG, and the 13 other LTG proteins in M. xanthus are not clear. Is *E. coli* MltG the closest homology to AgmT? Their relationships could be addressed with a phylogenetic tree and/or sequence alignments. Furthermore, are there other A-motility genes in proximity to agmT? Similarly, does agmT show specific co-occurrences with the other A-motility genes across genera/species?

We answered the first question in the Discussion (it was in the first Results section in the previous version), “Both *M. xanthus* AgmT and *E. coli* MltG belong to the YceG/MltG family, which is the first identified LTG family that is conserved in both Gram-negative and positive bacteria 25,41. About 70% of bacterial genomes, including firmicutes, proteobacteria, and actinobacteria, encode YceG/MltG domains25. The unique inner membrane localization of this family and the fact that AgmT is the only *M. xanthus* LTG that belongs to this family (Table S2) could partially explain why it is the only LTG that contributes to gliding motility”.

For the second, we added one sentence in the Results, “No other motility-related genes are found in the vicinity of *agmT*”.

For the third question, we do not believe a co-occurrence analysis is necessary. Because *M. xanthus* gliding is very unique but “about 70% of bacterial genomes, including firmicutes, proteobacteria, and actinobacteria, encode YceG/MltG domains25”, gliding should show no co-occurrence with the YceG/MltG family LTGs.

(4) Related to iii, what about the functional relationship of the endogenous 13 LTG genes? Although knockout mutants were shown to be motile, presumably because AgmT is present, can overexpression of them, similar to *E. coli* MltG, complement an agmT mutant? In other words, why does MltG complement and the endogenous LTG proteins appear not to be relevant?

We appreciate the reviewer for this question, which prompted us to think the uniqueness of AgmT more carefully. AgmT is unique for its inner-membrane localization, rather than activity. We answered this question in the discussion, “LTGs usually break glycan strands and produce unique anhydro caps on their ends40-44. However, because AgmT is the only LTGs that is required for gliding, it is not likely to facilitate bFAC assembly by generating such modification on glycan strands”. We then moved on to propose a possible mechanism, “*E. coli* MltG is a glycan terminase that controls the length of newly synthesized PG glycans25. Likewise, AgmT could generate short glycan strands and thus uniquely modify the overall structure of *M. xanthus* PG, such as producing small pores that retard and retain the inner subcomplexes of bFACs (Fig. 6). On the contrary, the *M. xanthus* mutants that lack active AgmT could produce PG with increased strain length, which blocks bFACs from binding to the cell wall and precludes stable bFAC assembly. However, it would be very difficult to demonstrate how glycan length affects the connection between bFACs and PG”.

(5) Based on Figure 2B, overexpression of MltG enhances A-motility compared to the parent strain and the agmT-PAmCh complemented strain, is this actually true? Showing expanded swarming colony phenotypes would help address this question.

We appreciate the reviewer for bringing up an important question. Prompted by this question, we analyzed the effects of MltG expression at the single-cell level. We found that “Consistent with its LTG activity, the expression of MltGEc restored gliding motility of the *ΔagmT pilA-* cells on both the colony (Fig. 2B) and single-cell (Fig. 2C, D) levels. Interestingly, in the absence of sodium vanillate, the leakage expression of MltGEc using the vanillate-inducible promoter was sufficient to compensate the loss of AgmT. A plausible explanation of this observation is that as *E. coli* grows much faster (generation time 20 - 30 min) than *M. xanthus* (generation time ~4 h), MltGEc could possess significantly higher LTG activity than AgmT. Induced by 200 μM sodium vanillate, the expression of MltGEc further but non significantly increased the velocity and gliding persistency (Fig. 2B-D). Importantly, the expression of MltGEc failed to restore gliding motility in the *agmTEAEA pilA* cells, even in the presence of 200 μM sodium vanillate (Fig. 2B). Consistent with the mecillinam resistance assay (Fig. 3C), this result suggests that AgmTEAEA still binds to PG and that in the absence of its LTG activity, AgmT does not anchor bFACs to PG”. These results are shown in the new panels C and D in Figure 2.

(6) Cell flexibility is correlated with gliding motility function in M. xanthus. Since AgmT has LTG activity, are agmT mutants less flexible than WT cells and is this the cause of their motility defect?

We appreciate the reviewer for bringing up an important question. We saw cells that lack AgmT making S-turns and U-turns frequently under microscope. We used a GRABS assay to quantify cell stiffness and found that neither the absence of AgmT nor the expression of MltGEc affect cell stiffness. We added this result in the manuscript, “The assembly of bFACs produces wave-like deformation on cell surface6,37, suggesting that their assembly may require a flexible PG layer2,6,11,12. As a major contributor to cell stiffness, PG flexibility affects the overall stiffness of cells38. To test the possibility that AgmT and MltGEc facilitate bFAC assembly by reducing PG stiffness, we adopted the GRABS assay38 to quantify if the lack of AgmT and the expression of MltGEc affects cell stiffness. To quantify changes in cell stiffness, we simultaneously measured the growth of the *pilA-*, *ΔagmT pilA-*, and *ΔagmT Pvan-MltGEc pilA-* (with 200 μM sodium vanillate) cells in a 1% agarose gel infused with CYE and liquid CYE and calculated the GRABS scores of the *ΔagmT pilA-*, and *ΔagmT Pvan-MltGEc pilA-* cells using the *pilA-* cells as the reference, where positive and negative GRABS scores indicate increased and decreased stiffness, respectively (see Materials and Methods and Ref38). The GRABS scores of the *ΔagmT pilA-*, and *ΔagmT Pvan-MltGEc pilA-* (with 200 μM sodium vanillate) cells were -0.06 ± 0.04 and -0.10 ± 0.07 (n = 4), respectively, indicating that neither AgmT nor MltGEc affects cell stiffness significantly. Whereas PG flexibility could still be essential for gliding, AgmT and MltGEc do not regulate bFAC assembly by modulating PG stiffness. Instead, these LTGs could connect bFACs to PG by generating structural features that are irrelevant to PG stiffness”.

**Reviewer #2 (Public Review):**
The manuscript by Carbo et al. reports a novel role for the MltG homolog AgmT in gliding motility in M. xanthus. The authors conclusively show that AgmT is a cell wall lytic enzyme (likely a lytic transglycosylase), its lytic activity is required for gliding motility, and that its activity is required for proper binding of a component of the motility apparatus to the cell wall. The data are generally well-controlled. The marked strength of the manuscript includes the detailed characterization of AgmT as a cell wall lytic enzyme, and the careful dissection of its role in motility. Using multiple lines of evidence, the authors conclusively show that AgmT does not directly associate with the motility complexes, but that instead its absence (or the overexpression of its active site mutant) results in the failure of focal adhesion complexes to properly interact with the cell wall.An interpretive weakness is the rather direct role attributed to AgmT in focal adhesion assembly. While their data clearly show that AgmT is important, it is unclear whether this is the direct consequence of AgmT somehow promoting bFAC binding to PG or just an indirect consequence of changed cell wall architecture without AgmT. In *E. coli*, an MltG mutant has increased PG strain length, suggesting that M. xanthus's PG architecture may likewise be compromised in a way that precludes AglR binding to the cell wall. However, this distinction would be very difficult to establish experimentally. MltG has been shown to associate with active cell wall synthesis in *E. coli* in the absence of protein-protein interactions, and one could envision a similar model in M. xanthus, where active cell wall synthesis is required for focal adhesion assembly, and MltG makes an important contribution to this process.

Based on the data that AgmT does not assemble into bFACs and that heterologous MltGEc substitutes *M. xanthus* AgmT in gliding, we believe that AgmT facilitates the proper assembly of bFACs indirectly. At the end of Introduction, we pointed out, “Hence, the LTG activity of AgmT anchors bFAC to PG, potentially by modifying PG structure”. Following the reviewer’s recommendation, we revised the Discussion to emphasize that AgmT facilitates proper bFAC assembly indirectly through its LTG activity. For the reviewer’s convenience, the revised paragraph is pasted here, with the changes highlighted in blue:

“It is surprising that AgmT itself does not assemble into bFACs and that MltGEc substitutes AgmT in gliding. Thus, rather than interacting with bFAC components directly and specifically, AgmT facilitates proper bFAC assembly indirectly through its LTG activity. LTGs usually break glycan strands and produce unique anhydro caps on their ends40-44. However, because AgmT is the only LTGs that is required for gliding, it is not likely to facilitate bFAC assembly by generating such modification on glycan strands. *E. coli* MltG is a glycan terminase that controls the length of newly synthesized PG glycans25. Likewise, AgmT could generate short glycan strands and thus uniquely modify the overall structure of *M. xanthus* PG, such as producing small pores that retard and retain the inner subcomplexes of bFACs (Fig. 6). On the contrary, the *M. xanthus* mutants that lack active AgmT could produce PG with increased strain length, which blocks bFACs from binding to the cell wall and precludes stable bFAC assembly. However, it would be very difficult to demonstrate how glycan length affects the connection between bFACs and PG”.

**Recommendations for the authors:**

**Reviewer #1 (Recommendations For The Authors):**
The last sentence of the Discussion implies that anchoring LTG (AgmT) in the inner membrane is important. I did not see this mentioned about AgmT. Does it contain an inner membrane anchoring domain? Along these lines, the AgmT and MltG proteins appear to be of different sizes (Figure 1A). Please clarify, perhaps including full-length sequence alignment and/or domain architecture for these proteins.

We revised the first paragraph in the Results and clarified, “Among these genes, agmT (ORF K1515_0491023) was predicted to encode an inner membrane protein with a single N-terminal transmembrane helix (residues 4 – 25) and a large “periplasmic solute-binding” domain22.”

We appreciate the reviewer for spotting the mistake in Fig. 2A. The *E. coli* MltG sequence shown in the alignment starts from residue 158, instead of 88. We have corrected this mistake in the figure. *M. xanthus* AgmT and *E. coli* MltG are of similar sizes, with 239 and 240 amino acids, respectively.

In Figure 3 legend, define D3.

The definition of D_3_ was added into the figure legend.

Figure 4A shows 100-frame composite micrographs, but no time interval between frames is given.

The imaging frequency, 10 Hz, was stated in the text. We also added this information into the figure legend.

Line 98, the term "Especially" does not flow well, change to "This includes the characteristic..." or similar.

We deleted “especially” from the sentence.

Line 179, "not" is not accurate, replace with "rarely."

Changed.

Line 188, add a qualifier, "proper" before "bFACs assembly."

Added.

Lines 196 and 202, provide the sizes of each protein in these fusion constructs.

We added these numbers to the figure legend.

In Figure 5A add arrows to identify each band. State in legend whether this is a denaturing gel, if so, why are AgmT-PAmCherry homodimers present?

Protein electrophoresis was done using SDS-PAGE. It is not unusual that some proteins, especially membrane proteins, are resistant to dissociation by SDS and appear as multimers in SDS-PAGE. The authors have seen this phenomenon repeatedly in both our experiments and the literature. Nevertheless, we clarified our experimental condition in the text, “Similar to many membrane proteins that resistant to dissociation by SDS34, immunoblot using an anti-mCherry antibody showed that AgmTPAmCherry accumulated in two bands in SDS-PAGE that corresponded to monomers and dimers of the full-length fusion protein, respectively (Fig. 5A)”.

A few examples for membrane proteins remaining as oligomers are listed in below:

Rath et al., 2009, PNAS 106: 1760-1765

Sulistijo *et al*., 2003, J Biol Chem 278: 51950-51956

Sukharev 2002, Biophy J 83: 290-298

Neumann *et al*., 1998, J Bacteriol 180: 3312-3316

Blakey *et al*., 2002, Biochem J 364: 527-535

Wegner and Jones, 1984, J Biol Chem 259: 1834-1841

Jiang *et al.*, 2002, Nature 417: 515-522

Heginbotham and Miller, 1997, Biochem 36: 10335-10342

Gentile et al., 2002, J Biol Chem 277: 44050-44060

Line 207, "near evenly along cell bodies" does not seem consistent with Figure 5B as there looks to be an enrichment of AgmT at cell poles.

We have replaced panel 5B with more typical images. Due to the shape difference between cell poles and the cylindrical nonpolar regions, many surface-associated proteins could appear “enriched” at cell poles. This effect was very obvious in Fig. 5B, possibly due to the unevenness of the agar surface. We examined our data carefully and did not find significant polar enrichment. Compared to AglZ that significantly enriches at poles and forms evenly-spaced clusters along the cell body, the localization of AgmT is completely different.

Lines 252 and 260, change "Fig. 5B" to "Fig. 5C."

We apologize for these mistakes. They have been corrected.

Line 266, insert "the" before "cell envelope."

Added.

Line 278, insert "presumably" between "AgmT generates (small openings)"

Corrected.

**Reviewer #2 (Recommendations For The Authors):**
- Major comment: I would rephrase conclusions regarding a direct role of AgmT in focal adhesion assembly since these data are indirect (AglR binding to the cell wall is reduced in the absence of AgmT - this could also be interpreted as the absence of AgmT causing altered cell wall architecture that precludes AglR binding). Example: I don't think the data support line 222 "AgmT connects bFACs to PG", perhaps rephrased to accommodate more agnostic explanations. Likewise, line 308 states that MltG has been "adopted" by the gliding motility machinery. This conclusion cannot be drawn from the data presented.

We agree with the reviewer that the conclusions should be stated precisely. At the end of Introduction, we pointed out, “Hence, the LTG activity of AgmT anchors bFAC to PG, potentially by modifying PG structure”. Following the reviewer’s recommendation, we revised the Discussion to emphasize that AgmT facilitates bFAC assembly indirectly through its LTG activity. For the reviewer’s convenience, the revised paragraph is pasted here, with the changes highlighted in blue:

“It is surprising that AgmT itself does not assemble into bFACs and that MltGEc substitutes AgmT in gliding. Thus, rather than interacting with bFAC components directly and specifically, AgmT facilitates proper bFAC assembly indirectly through its LTG activity. LTGs usually break glycan strands and produce unique anhydro caps on their ends40-44. However, because AgmT is the only LTGs that is required for gliding, it is not likely to facilitate bFAC assembly by generating such modification on glycan strands. *E. coli* MltG is a glycan terminase that controls the length of newly synthesized PG glycans25. Likewise, AgmT could generate short glycan strands and thus uniquely modify the overall structure of *M. xanthus* PG, such as producing small pores that retard and retain the inner subcomplexes of bFACs (Fig. 6). On the contrary, the *M. xanthus* mutants that lack active AgmT could produce PG with increased strain length, which blocks bFACs from binding to the cell wall and precludes stable bFAC assembly. However, it would be very difficult to demonstrate how glycan length affects the connection between bFACs and PG”.

However, we believe that the conclusion that “AgmT connects bFACs to PG" still stands true. Although AgmT is not likely to interact with the gliding machinery directly, its activity does increase the binding between bFACs and PG.

We agree with the reviewer that “adopt” may not be the best word to describe AgmT’s function in gliding. In the revised manuscript, we changed the phrase to “contributes to gliding motility”.

- Line 35: define "bFAC" at first use.

Fixed.

- Figure 2: Mention in the caption why the pilA mutation is significant. Also, make more clear what one is supposed to see. You could include an arrow showing motile cells extruding from the colony edge, and mark + label the edge of the colony.

Following the reviewer’s recommendations, we described the motility phenotypes in detail in the main text, “On a 1.5% agar surface, the *pilA-* cells moved away from colony edges both as individuals and in “flare-like” cell groups, indicating that they were still motile with gliding motility. In contrast, the *∆aglR pilA-* cells that lack an essential component in the gliding motor, were unable to move outward from the colony edge and thus formed sharp colony edges. Similarly, the *∆agmT pilA-* cells also formed sharp colony edges, indicating that they could not move efficiently with gliding (Fig. 2B)”.

We also added a schematic block into panel B and two sentences into the legend, “To eliminate S-motility, we further knocked out the *pilA* gene that encodes pilin for type IV pilus. Cells that move by gliding are able to move away from colony edges.”

- Figure 3 caption. Mecillinam concentration should presumably be µg/mL, not g/mL?

Also, remove the ".van,." in the second to last line.

We apologize for these mistakes. We have corrected them in the figure legend.

- Line 212 - at this point in the manuscript, the fact that AgmT likely does not assemble into bFACs is quite well established, so I would start this paragraph with something like "As an additional test, we...".

Revised as the reviewer recommended.

- Figure 5C - this assay needs a protein loading control. How about whole-cell AglR before pelleting PG?

We do have a whole-cell loading control, which we have added into the revised figure.

- Figure 5A - how are the dimers visible? Is this a native gel? If so, please add to the Methods section (I would find information on Western Blot there, but not on gel electrophoresis).

Protein electrophoresis was done using SDS-PAGE. It is not unusual that some proteins, especially membrane proteins, are resistant to dissociation by SDS and appear as multimers in SDS-PAGE. The authors have seen this phenomenon repeatedly in both our experiments and the literature. Nevertheless, we clarified our experimental condition in the text, “Similar to many membrane proteins that resistant to dissociation by SDS34, immunoblot using an anti-mCherry antibody showed that AgmTPAmCherry accumulated in two bands in SDS-PAGE that corresponded to monomers and dimers of the full-length fusion protein, respectively (Fig. 5A)”.

A few examples for membrane proteins remaining as oligomers are listed in below:

Rath et al., 2009, PNAS 106: 1760-1765

Sulistijo *et al*., 2003, J Biol Chem 278: 51950-51956

Sukharev 2002, Biophy J 83: 290-298

Neumann *et al*., 1998, J Bacteriol 180: 3312-3316

Blakey *et al*., 2002, Biochem J 364: 527-535

Wegner and Jones, 1984, J Biol Chem 259: 1834-1841

Jiang *et al.*, 2002, Nature 417: 515-522

Heginbotham and Miller, 1997, Biochem 36: 10335-10342

Gentile et al., 2002, J Biol Chem 277: 44050-44060